# Rotavirus Seasonality: An Application of Singular Spectrum Analysis and Polyharmonic Modeling

**DOI:** 10.3390/ijerph16224309

**Published:** 2019-11-06

**Authors:** Olga K. Alsova, Valery B. Loktev, Elena N. Naumova

**Affiliations:** 1Novosibirsk State Technical University, Novosibirsk 630073, Russia; alsova@corp.nstu.ru; 2Institute of Cytology and Genetics, Siberian Branch of the Russian Academy of Sciences, Novosibirsk 630090, Russia; valeryloktev@gmail.com; 3State Research Center for Virology and Biotechnology “Vector”, Koltsovo, Novosibirsk Region 630559, Russia; 4Friedman School of Nutrition Science and Policy, Tufts University, Boston, MA 02111, USA

**Keywords:** time series analysis, singular spectrum analysis, periodogram spectral analysis, Poisson polyharmonic regression model, rotavirus, seasonality, ambient temperature, cold climate, Russia

## Abstract

The dynamics of many viral infections, including rotaviral infections (RIs), are known to have a complex non-linear, non-stationary structure with strong seasonality indicative of virus and host sensitivity to environmental conditions. However, analytical tools suitable for the identification of seasonal peaks are limited. We introduced a two-step procedure to determine seasonal patterns in RI and examined the relationship between daily rates of rotaviral infection and ambient temperature in cold climates in three Russian cities: Chelyabinsk, Yekaterinburg, and Barnaul from 2005 to 2011. We described the structure of temporal variations using a new class of singular spectral analysis (SSA) models based on the “Caterpillar” algorithm. We then fitted Poisson polyharmonic regression (PPHR) models and examined the relationship between daily RI rates and ambient temperature. In SSA models, RI rates reached their seasonal peaks around 24 February, 5 March, and 12 March (i.e., the 55.17 ± 3.21, 64.17 ± 5.12, and 71.11 ± 7.48 day of the year) in Chelyabinsk, Yekaterinburg, and Barnaul, respectively. Yet, in all three cities, the minimum temperature was observed, on average, to be on 15 January, which translates to a lag between the peak in disease incidence and time of temperature minimum of 38–40 days for Chelyabinsk, 45–49 days in Yekaterinburg, and 56–59 days in Barnaul. The proposed approach takes advantage of an accurate description of the time series data offered by the SSA-model coupled with a straightforward interpretation of the PPHR model. By better tailoring analytical methodology to estimate seasonal features and understand the relationships between infection and environmental conditions, regional and global disease forecasting can be further improved.

## 1. Introduction

Global surveillance for major infectious diseases, such as influenza and rotaviral infection—one of the leading causes of mortality in both young children and older adults, increases the demand for reliable tools to streamline data analysis, visualization, and interpretation of surveillance records. The dynamics of many viral infections is known to have a complex non-linear, non-stationary structure due to the multiple routes of transmissions and penitential amplifiers affecting host health. Viral rotaviral infections (RIs) exhibit strong seasonality, characterized by marked changes in disease incidence over a course of a calendar year when relatively short periods of high incidence alternate with prolong periods of low incidence, indicative of virus sensitivity to environmental conditions [1,2,3,4,5]. To properly guide public health intervention programs, it is important to carefully describe the dynamics of infection and provide reliable estimates of seasonal characteristics. Furthermore, understanding the relationship between infection and environmental temperature is essential for disease forecasting on global and local scales. 

However, analytical tools suitable for identification of seasonal peaks along with their uncertainty metrics are limited and rarely used in a systematic manner. In surveillance reports worldwide, seasonal peaks are often presented by the month with the highest number of cases, which is insufficient to accurately compare seasonal peaks across populations, geographic locations, and time periods. Yet, if the incidence of an infectious disease observed in a specific population is tracked in a systematic manner and is expressed in a proper chronological order at high temporal resolution, researchers can apply modern sophisticated methodology to conduct in-depth data analysis. In other words, if we assemble cases of health outcomes as they occur in time, aggregate these events with respect to time units, say into daily or weekly cases, and compile these counts or rates into so-called time series, we will be able to apply analytical methods suitable for time series data. 

Infections that are highly contagious often exhibit sharp spikes in time series of disease incidence. Infections that are highly seasonal tend to peak at the same time of the year. Methods that are most critical in describing seasonal features include models for: 1) adequate selection of a form for a long-term trend, 2) identification of a cycle length or period, and 3) detection of a seasonal peak and amplitude along with the relevant measures of uncertainty. Thus, the models describing time series of disease records have to take into account a potentially complex structure that is non-linear and non-stationary. Non-stationarity and non-linearity are likely to be present when an infection is sensitive to environmental conditions governing pathogen’s transmission and host health.

Rotavirus, one of the most common pediatric diarrheal diseases, typically peaks in cold and dry weather in both temperate and tropical climates [1,2,3,4,5,6,7,8,9,10,11,12]. However, the seasonal pattern and strength of the association between the incidence of rotaviral infection and ambient temperature, and the sensitivity of a seasonal peak of rotavirus to temperature vary. Some studies demonstrated a delay when the seasonal peak occurred 1–2 months after a nadir in ambient temperature [3,6,13]. It is not clear how stable the seasonal peak is and how strong the relationship between rotavirus infections and ambient temperature is in cold climates. The relationship between temperature and seasonal infection, including RI, is difficult to compare across the studies due to the marked differences in the methods applied to assess such a relationship, the differences in climatic and weather conditions, heterogenicity of RI infections across populations, degree of vaccination coverage, and many other factors. Studies that compare models to capture sensitivity of a seasonal peak of rotavirus to temperature are rare. The accuracy of assessing relationships between temperature and infection often depends on how values are aggregated across time—the more refined the time units, the higher the expected accuracy. In other words, daily time series is superior to weekly and monthly time series. Most of the studies used weekly or monthly records and, thus, created additional ambiguity with selecting the most appropriate metrics for ambient temperature, which could take a variety of forms [3,7,8,9,10,14]. Some studies assessed the relationship with temperature by simply comparing cold and warm seasons [9,10,14]. In some studies, peaks of infections were presented as the month with highest counts [6], with occasional reporting of two seasonal peaks in RI [3,14]. Our understanding of what infection seasonality represents is currently limited by a lack of standardization of methodology for in-depth analysis of temporal variations in infections with marked seasonality and potential environmental factors that may act as proxies for seasonal change. 

Documenting how seasonal features change over time is important in assessing the effect of broadly implemented vaccination strategies. With the growing potential for wide implementation of vaccination against rotavirus [15,16,17,18,19], there is increasing evidence that the seasonal pattern changes due to the fact of vaccination; for example, in Brazil the seasonal oscillations became less pronounced [16]. With the ongoing climatic changes, it could be expected that the seasonality of rotavirus might change as well. Thus, there is a clear need for establishing a baseline temporal pattern in RI and its sensitivity to environmental temperature. Such description requires a solid methodology for assessing systematic patterns, most importantly the seasonal fluctuations. The methodology has to provide parameters that can unambiguously characterize the temporal behavior in the incidence of infection. Specifically, researchers should be able to estimate seasonal characteristics, like peak timing, along with a proper measure of uncertainty. This methodology requires high-quality primary medical data collection where each record has a timestamp. These time-referenced records can be arranged in chronological order enabling time series analysis. Novel electronic health record systems that monitor and track diseases over time are offering such data streams and are likely to benefit the most from these novel analytical advancements.

In our earlier work, we demonstrated the use of harmonic regression models—models that explicitly use sine and cosine periodic functions to imitate seasonal variations in the time series of disease incidence [20,21,22,23,24,25]. The results of the harmonic regression model produce the estimates of parameters for sine and cosine terms with proper specification and, if any of two parameters is statistically significant, one may suspect that the temporal variations are periodic within a one-year cycle or seasonal. We also developed the δ-method to estimate the average peak timing and peak intensity [20,21,22,23,24,25]. This method converts the results of a harmonic regression model, specifically the regression coefficients of the sine and cosine periodic terms, into time units, such as days, weeks or months, to accurately define when the disease incidence reaches its annual peak or nadir. This method also provides uncertainty measures that provide a sense of how much this peak timing may vary or how stable it is. In situations when disease incidence exhibits a relatively stable seasonal curve with one well-defined peak, this model explains a large fraction of variability and provides reliable estimates with the straightforward interpretation. However, the main drawback of this approach is that the harmonic model is symmetrical in nature—it assumes the same rate of increase and decrease in disease incidence from nadir to peak and vice versa. 

Polyharmonic regression models in epidemiological research act as a simplified version a classical spectral analysis (SA) [26]. In fact, the classical SA applied to time series of disease incidence aims to find a recurrent periodic pattern—a pattern that repeats itself over time, for example, a summer increase in enteric infections observed every year—and to determine the relative importance of periodic oscillations of various nature such as bi-annual or weekly variations. This method does not require a priori knowledge of model structure. However, spectral analysis has high demand on the length of time series, requires preliminary detrending, and the interpretation of models is complex. Recent development of singular spectrum analysis (SSA) is a promising alternative to SA and is capable of substantially reducing noise, dealing with trend components, and revealing the temporal structure of the data without preliminary manipulation [27,28]. The method of SSA is a multidimensional analogue of a principal component analysis adapted to time series. The basic idea of the SSA is to transform a one-dimensional time series into multi-dimensional trajectories using principal component analysis (singular value decomposition) and reconstruction (approximation) of a number of selected principal components. While the SSA has high demand on the time series length and lacks parametric description, the SSA-based models are flexible and allow to recreate asymmetric shapes of a seasonal curve and, thus, better predict seasonal peaks compared to harmonic models. To the best of our knowledge, the use of SSA models in the analysis of epidemiological data has never been done before. In this study we took the advantage of both approaches: the accuracy and flexibility of SSA and the easy and interpretability of polyharmonic regression. 

In the present study, we developed a two-step modeling procedure to take advantage of both approaches. First, we identified periodicity in the observed data and, then, using this information, built models to determine and compare seasonal characteristics. We demonstrated the approach by examining the relationship between daily time series of RI rates and ambient temperature in three Russian cities from 2005 to 2011 using SSA and Poisson polyharmonic regression (PPHR) models. We identified the structure of temporal variation in daily time series of RI rates and temperature using the SSA model—a new class of spectral analysis models. We then applied PPHR models to examine the relationship between daily time series of RI and ambient temperature and estimate seasonal peaks. We provide the models’ assessments and an interpretation of modeling results. We conclude this communication with the discussion of main findings, limitations, and further research directions. 

## 2. Data and Methods 

We abstracted all fully anonymized laboratory-confirmed cases of rotavirus (A08.0, ICD-10), representing rotaviral infection in three Russian cities: 12,423 cases in Chelyabinsk, 13,342 cases in Yekaterinburg, and 3307 cases in Barnaul as recorded by the Climate–Water–Diseases–Infections (CliWaDIn) database version 1.0 [29,30]. Using the date of testing, we created daily time series of cases for each city. To compare rotavirus incidence between cities, the time series of cases were converted into time series of daily rates per 1 million persons. As denominators, we used the annual population for each city and each year: for Chelyabinsk—from 1 January 2005 to 30 June 2011 (2372 study days); for Yekaterinburg—from 1 January 2005 to 31 July 2011 (2403 study days); and for Barnaul—from 1 January 2006 to 31 December 2011 (2191 study days) as shown in Table 1. 

Annual demographic data were obtained from the official Russian census website [31]. Daily average temperature records were obtained from the city public sites [32] and then stored, curated, and maintained by the CliWaDIn database [29].

Two cities, Chelyabinsk and Yekaterinburg, are located relatively close to each other (~193 km apart) and have over 1 million residents. Barnaul is about half the size and is 1460 km and 1520 km away from Chelyabinsk and Yekaterinburg, respectively (Figure 1). All three cities shared similar climate conditions according to the Köppen–Geiger climate classification system [33]. The Köppen climate classification divides climates into five main climate groups: A (tropical), B (dry), C (temperate), D (continental), and E (polar), with each group being divided based on seasonal precipitation and temperature patterns, also represented by a letter.

### 2.1. Exploratory Analysis 

We examined the suitability of a Poisson distribution for daily RI rates based on the distributional shape, the coefficients of skewness and kurtosis, and the Kolmogorov–Smirnov test. We applied the periodogram spectral analysis to identify temporal components and seasonal periodicity.

### 2.2. SSA Method

The basic SSA algorithm decomposes an original non-zero time series into four main components: trend, seasonal component, additional periodic components, and noise. An original non-zero time series, $Y_{t}$ is a series $Y_{t}=\left(y_{1},\ldots ,y_{N}\right)$ of length *N* (*N* > 2) and with at least one value of *i* when $y_{i}\neq 0$. In the presented analysis, we constructed the SSA model (Model A) of the time series of daily RI rates using the SSA algorithm, which consists of four steps: embedding, singular value decomposition (SVD), eigentriple grouping, and diagonal averaging, as described below.

Step 1: Embedding. The main feature of the SSA algorithm is the window of length (*L*) 1<L<N. To perform the first step of embedding, we mapped the original time series into a sequence of *K = N − L +* 1 lagged vectors of size *L*:(1)$$Y_{i}={\left(y_{i},\ldots ,y_{i+L-1}\right)}^{T},1\leq i\leq K,$$
to form the trajectory matrix ***Y*** of the series $Y_{t}$, defined as:(2)$$Y=\left[Y_{1}:\ldots :Y_{K}\right]={\left(y_{ij}\right)}_{i,j=1}^{L,K}=\left(\begin{array}{cccc} y_{1} & y_{2} & \ldots & y_{K}\\ y_{2} & y_{3} & \ldots & y_{K+1}\\ \ldots & \ldots & \ldots & \ldots \\ y_{L} & y_{L+1} & \ldots & y_{N} \end{array}\right)$$

Step 2: Singular value decomposition. At the second step, we performed the SVD procedure of the trajectory matrix ***Y***: (3)$$Y=Y_{1}+Y_{2}\;+\ldots +\;Y_{d},\;\;Y_{i}=\sqrt{\lambda_{i}}U_{i}V_{i}^{T},\;\;i=1,\ldots ,d$$ where $\left\{U_{1},\;\ldots ,U_{d}\right\}$ is the corresponding orthonormal system of the eigenvectors of the matrix $S=YY^{T}$ with $\lambda_{1}\geq \lambda_{2}\geq \ldots \geq \lambda_{d}>0$ as the ordered non-zero eigenvalues, and $V_{i}=Y^{T}U_{i}/\sqrt{\lambda_{i}}$ are the factoring vectors. The collection $\left(\sqrt{\lambda_{i}},U_{i},V_{i}\right)$ is called the *i*-th eigentriple of the SVD, so all information about each of the components $Y_{i}$ is contained in the *i*-th eigentriple.

Step 3: Eigentriple grouping. The eigentriple grouping procedure partitions the set of indices {1,…,d} into *m* disjoint subsets $I_{1},\ldots ,I_{m}$, where $I=\left\{i_{1},\ldots ,i_{p}\right\}$. The resultant matrix $Y_{I}$ that correspond to the group *I* is defined as $Y_{I}=Y_{i_{1}}+\ldots +Y_{i_{p}}$. When the resultant matrices are computed for the groups $I=I_{1},\ldots ,I_{m}$ , the expansion (2) leads to the decomposition:(4)$$Y=Y_{I_{1}}+Y_{I_{2}}+\ldots +Y_{I_{m}}$$

Step 4: Diagonal averaging. At this step, we transformed each matrix $Y_{I_{j}}$ of the grouped decomposition (3) into a new series of length *N*. Diagonal averaging applied to a resultant matrix $Y_{I_{k}}$ produces a reconstructed series ${\tilde{Y}}^{\left(k\right)}=\left({\tilde{y}}_{1}^{\left(k\right)},\ldots ,{\tilde{y}}_{N}^{\left(k\right)}\right)$. Therefore, the initial series $\left(y_{1},\ldots ,y_{N}\right)$ was decomposed into a sum of *m* reconstructed series:(5)$$y_{n}=\overset{m}{\underset{k=1}{\sum }}{\tilde{y}}_{n}^{\left(k\right)},n=1,2,\ldots ,N$$

The process of finding the complete decomposition of the original time series is equivalent to the identification of eigentriple of SVD of the trajectory matrix Y, corresponding to the trend, periodic components, and noise and their grouping by the component types. 

We identified the temporal components for daily RI visually based on the analysis of graphs of eigenvectors. In general, eigenvectors have the same form as a component of the original series, to which they correspond. An eigenvector corresponding to the trend changes slowly and describes a long-term trend in the time series. The seasonal component with a period *T*, is defined by a pair of eigenvectors. In an ideal situation, this pair follow sine and cosine functions with period T and identical amplitude and phase. Thus, a two-dimensional diagram constructed on a plane of neighboring eigenvectors should have a spiral form. The number of turns of the resulted spiral indicates the period of a harmonic component. Finally, a period of a harmonic component is a ratio of the number of components in the expansion (the window length) to the number of spiral turns. 

### 2.3. Poisson Polyharmonic Regression (PPHR) Model

To describe the temporal patterns in the time series of daily RI rates and to estimate peak timing, we modeled the random sequences, $Y_{t}$, in the form of additive mixtures of harmonic components with periods corresponded to the number of days per year using a stationary regression with Poisson-distributed discrete noise, $e_{t}$ where $E\left\{e_{t}\right\}=0$ and $Var\left\{e_{t}\right\}=\sigma^{2}$. The general form of a PPHR model is:(6)$$Y_{t}=\exp(d_{0}+\overset{k}{\underset{i=1}{\sum }}\left(\alpha_{i}\sin(2\pi i\omega t\right)+\beta_{i}\cos(2\pi i\omega t))+e_{t})$$ where $Y_{t}$ is the outcome interest at time *t*; *t = 1, 2, …, N* is time in days, where *N* is the number of days in a time series; $d_{0}$ is an intercept and the $\exp\{d_{0}\}$ reflects mean daily values over the study period. The periodic *sin* and *cos* components have a frequency of ω *=* 1/365.25; $\alpha_{i},\beta_{i},i=1,\ldots ,k$ are the estimated parameters of the model (or regression coefficients) for *k*—number of the periodic components. This model can also contain the trend terms reflecting the general dynamics of change during the observation period. 

From the general form (5), we selected the components that were most reasonable to describe the annual and semi-annual oscillations based on the periodogram spectral analysis and the results of SSA modeling (a detailed explanation of how SSA modeling was used to build PPHR model is provided in Section 3). Thus, we built the PPHR for the daily rates with the following structure (Model B):(7)$$Y_{t}=\exp(d_{0}+d_{1}t+d_{2}t^{2}+d_{3}t^{3}+\alpha_{1}sin\left(2\pi \omega t\right)+\beta_{1}\cos\left(2\pi \omega t\right)+\alpha_{2}\sin(4\pi \omega t)+\beta_{2}\cos(4\pi \omega t)+e_{t})$$ where $d_{i},\;i=0,\ldots ,3;\alpha_{1},\;\alpha_{2},\;\beta_{1},\;\;\beta_{2}$, the estimated parameters of the model. The seasonal terms: sin(2πωt) and cos(2πωt) represent the annual oscillation (when one seasonal peak is expected); the seasonal terms: sin(4πωt) and cos(4πωt) represent the semi-annual oscillation (when two seasonal peaks are likely); the function of trend is the third-degree polynomial to account for a potential overall slow non-linear change in RI over the study period. To ease the interpretation of model parameters, we also explored a simplified version of Model B, Model B’ with one seasonal term and one linear trend term: (8)$$Y_{t}=\exp(d_{0}+d_{1}t+\alpha_{1}\sin(2\pi \omega t)+\beta_{1}\cos(2\pi \omega t)+e_{t})$$

We compared Model B and B’ to understand the improvements in model accuracy by considering additional terms.

### 2.4. Analysis of the Relationship between Daily RI Rates and Ambient Temperature

To examine the relationship between daily RI rates and ambient temperature, we expanded PPHR model (Models B and B’) by including two terms for ambient temperature: one term ($\gamma_{1}z_{t-1})$ for an immediate effect with 1-day lag (*t-1*) and the second term ($\gamma_{2}z_{t-l}),$ for a lagged effect with a variable lag (*t-l*). Thus, Models C and its simplified version C’ have the structure:(9)$$Y_{t}=\exp(d_{0}+d_{1}t+d_{2}t^{2}+d_{3}t^{3}+\alpha_{1}\sin(2\pi \omega t)+\beta_{1}\cos(2\pi \omega t)+\alpha_{2}\sin(4\pi \omega t)+\beta_{2}\cos(4\pi \omega t)\;+\gamma_{1}z_{t-1}+\gamma_{2}z_{t-l}+e_{t})$$
and:
(10)$$Y_{t}=\exp(d_{0}+d_{1}t+\alpha_{1}\sin(2\pi \omega t)+\beta_{1}\cos(2\pi \omega t)+\gamma_{1}z_{t-1}+\gamma_{2}z_{t-l}+e_{t})$$ where $\gamma_{1},\gamma_{2}$ are the estimated parameters for the temperature terms; $z_{t-1}$ is the value of the average temperature on day *t*-1; $z_{t-l}$ is the value of the average temperature on day *t-l*; *l* is the lag in days between the time of disease incidence maximum and the time of temperature minimum. The inclusion of two terms for temperature is justified by a potential difference between a seasonal peak in RI and a seasonal nadir in temperature. If such difference is detected the two terms should reflect the immediate and the lagged effects. 

To estimate this lag *l* and to determine the seasonal nadir in the daily time series of temperature in each city, we used a harmonic regression model with a Gaussian distribution for ambient temperature $Z_{t}$ as the outcome variable, Model D: (11)$$Z_{t}=d_{0}+\alpha_{1}\sin(2\pi \omega t)+\beta_{1}\cos(2\pi \omega t)+e_{t}$$ where $d_{0}$ is the mean of the ambient temperature over the study period; $\alpha_{1},\beta_{1}$ are the estimated parameters of the model for two harmonic terms as in Models B’ and C’; ω=1/365.25 is the time series frequency; $e_{t}$ is the model residuals. In cold climates, the average ambient temperature exhibits only one seasonal peak and one season nadir, thus two harmonic terms should be sufficient. 

We estimated the parameters of Model C’ and determined the statistical significance of the contribution of the temperature component in explaining the variance of the time series of daily RI rates. We also estimated the relative risk (RR) of daily RI, associated with immediate (RR1) and lagged (RR2) effects in temperature associated with 1 °C, as:(12)$$RR_{1}=\exp\left\{\gamma_{1}\right\}\;\mathrm{and}\;RR_{2}=\exp\left\{\gamma_{2}\right\}$$ and their lower and upper bounds of the confidence intervals (95%CI), as: (13)$$CI_{RR1}=\exp\left\{\gamma_{1}\pm 1.96se\right\}\;\mathrm{and}\;CI_{RR2}=\exp\left\{\gamma_{2}\pm 1.96se\right\}$$

To estimate the model parameters obtained with PPHR approaches, we used the iterative Gauss–Newton least squares method [34]. We plotted the values of RI rates estimated from Models A, B’, and C’. Finally, we estimated the average seasonal peaks of disease incidence from Model A, Model B’, and Model C’ and estimated the difference (lag) between the peak timing in disease incidence and the time when temperature reaches its minimum.

The quality of fit for the constructed models was evaluated by the coefficient of determination *D*, reflecting the overall proportion of variance in $Y_{t}$ explained by the model components and the proximity to the Poisson-distributed residuals, $e_{t}$, having the minimum values of the statistical characteristics (mean, median, minimum, maximum, standard deviation, and mean absolute error (MAE)). All analysis was performed using Statistica 12.0 (StatSoft Inc., Palo Alto, CA, USA) statistical software. Codes for SSA models [35] were based on the “Caterpillar” SSA algorithm and can be requested from the author (O.K.A.). The compiled dataset containing daily time series of RI rates and temperature is provided in the Appendix A. 

## 3. Results

The daily time series of RI rates (estimated from reported daily counts and interpolated city population) and average temperature for three cities: Chelyabinsk, Yekaterinburg, and Barnaul, 2005–2011, are shown in Figure 2. All three cities shared similar temperate climatic conditions with cold winters and relatively warm summers (Table 2). Overall, the RI rate in Barnaul was twice as lower as compared to the two other cities. For all cities, RI rates showed a slow upward trend and a well-pronounced recurrent increase with high spikes in the wintertime, typically after the nadir in temperature and prolonged periods with low rates. In some years, there were irregular spikes in RI during summer months. As part of an exploratory analysis, we identified the distributional and temporal structure of time series of daily RI rates. The distributions of RI rates appeared to be right-skewed based on high values of skewness and kurtosis and driven by the seasonal winter spikes when daily maximums were ~10 fold higher than the annual median values (Table 2 and Figure 2), although the Kolmogorov–Smirnov tests rejected the Poisson assumption due the high contribution of days with low rates, shown by the right shift of the fitted distribution as compared to the actual values. 

Using the periodogram spectral analysis for RI rates, we determined that statistically significant periodogram peaks correspond to periods of 343.43 and 184.92 days in Yekaterinburg; 338.86 and 182.46 days in Chelyabinsk; 365.0 and 182.5 days in Barnaul (Figure 3). These results indicate that, in the studied time series, there were frequencies close to the annual cycle and half year cycle—in other words, the annual and semi-annual oscillations. Periodogram results for temperature showed a well-defined single peak at 365.0 day for Barnaul and at 365.1 day for Yekaterinburg and Chelyabinsk (Figure 3).

To further identify and describe the periodicity of time series of daily RI rates, we applied the SSA modeling procedure. The analysis was conducted according to the steps described in Section 2.2. The window length was selected to be equal to the number of components of SVD and was set as one annual period to achieve high accuracy. We identified three types of components: trend, seasonal components, and noise (the remainder). Figure 4 shows the presence of trend and seasonality based on eigenvectors of the first five eigentriples of the SVD for the time series of daily RI rates in three cities.

As shown on Figure 4a,d,g, the first eigenvector (1-EV) changed slowly with respect to the lag; it describes the non-linear trend in RI rates, suggesting the need for considering additional trend terms in the harmonic regression models. Four other eigenvectors (2–5 EVs) exhibited regular periodic behavior, suggesting the need for seasonal terms. The second and third rows of Figure 4 show two-dimensional diagrams of the neighboring eigenvectors ordered by eigenvalues in eigentriples. The relationship among the eigenvectors formed tight spirals with one (for 2–3 EVs) or two (4–5 EVs) coils. The number of coils indicates the period of a harmonic component. Therefore, the single coil for 2–3 EV components described an annual periodicity equal to 365 days, warranting an annual seasonal term in the harmonic regression models. The dual coils for 4–5 EVs components described a six-month interval (182 days) oscillations in the time series, suggesting an additional term for semi-annual oscillations. For all three cities, the results of SSA models (Model A) resembled the estimate obtained from spectral analysis, with the annual and semi-annual periods of 365.25 and 182.6 days (counting the leap year), respectively. 

Next, we constructed PPHR models informed by the finding from the spectral analysis and SSA modeling. Table 3 shows the statistical characteristics of the accuracy of the model identification based on the residuals of Models A, B, and B’. The variance explained by Model A was consistently higher as compared to Model B, by ~3–4%. As expected, for the simplified Model B’, the difference increased by ~5–7.5% for all three cities: 52.7% versus 47.75%; 62.77% versus 57.45%; and 43.06% versus 35.54% for Chelyabinsk, Yekaterinburg, and Barnaul, respectively. The results suggest that the approximation of the time series of daily RI rates with Model B’ was relatively crude and had limited ability to pick up the high values yet was sufficient to describe the seasonality. The inclusion of the ambient temperature predictors to Models B and B’ offered a slight improvement but did not reach the fit of Model A. The harmonic regression of Model D applied to the daily time series of the temperature offered a reasonably good fit. 

The contribution of trend and seasonal components in explaining the variance of time series of daily RI rates differed by model and by city. In general, the trend and annual cycle for all three cities explained most of the outcome variance. There was a significant positive trend in the RI rates in all three cities. Among PPHR models, Model B offered the most comprehensive description of temporal components, including non-linear variations in trend and the semi-annual seasonal component (Table 4). The contribution of the trend and the annual period to the explained variance of the time series of daily RI rates depended on the city: in Chelyabinsk, these two components explained 11.48% and 36.73%, in Yekaterinburg 11.67% and 46.49%, and in Barnaul 21.46% and 18.51%. The contribution of the second peak of the seasonal (semi-annual period) had no statistical significance; therefore, its identification and description were not carried out. The coefficients of the trend function corresponding to the second and third degrees were also not statistically significant. These results provide justification for the simplified version (Model B’), which was then extended to include the terms related to temperature (Models C and C’). In order to select a proper lag for temperature in Models C and C’, we examined the seasonal peaks for ambient temperature in the three cities using Model D (Table 3). In all three cities, the minimum temperature was observed on average on the 15 of January. 

The variance explained by Model C’ was similar to the values of Model B’ (Table 3) for all three cities. The inclusion of the temperature component with the selected lag increased the accuracy of the identification of the time series of daily RI rates. The immediate effect of temperature (RR1) was non-significant (*p* > 0.05), yet the lagged effect (RR2) was statistically significant (*p* < 0.05) indicating the low ambient temperature at its nadir correlated with high RI rates. The estimates of the relative risks (RR) associated with an increase by 1 °C for the lagged temperature effect were significant for all cities, as shown by the values of lower and upper bounds of the 95% confidence interval (LCI_RR2_ and UCI_RR2_). However, the percentage of the variance of the time series explained by the temperature components was low: 0.01%–0.04% (immediate effect) and 0.26%–0.49% (lagged effect), depending on the city (Table 4). 

Next, the time series of daily RI rates were reconstructed by models. The results of the select comprehensive Models A, B’, and C’ are shown in Figure 5. As expected, all models depicted the general seasonal nature of RI and predicted from 36% to 59% of variability. Model A obtained via SSA modeling provided the most refined structure of the studied time series of RI rates. In addition to a well-pronounced winter peak, Model A revealed irregularities and unusual summertime peaks. As expected, the two harmonic regression models, Models B’ and C’, identified a temporal structure with one seasonal peak for daily RI rates in three cities similarly to Model A. Because Model C’ included a term for ambient temperature, it depicted variations in RI rates due to the ambient temperature, mostly noted when RI rate reached its peaks. Thus, Model C’ demonstrated a modest contribution of temperature in predicting RI rates.

Models B’ and C’ are simplified versions of Models B and C and allow to ease the interpretation of model results. Peak timing estimates provided by Models A, B’ and C’ are shown in Table 5. Model B’ offered a substantially narrower and the least conservative estimate range. Based on the presented models RI peaked first in Chelyabinsk during 22−24 February, then in Yekaterinburg from 1−5 of March, and finally in Barnaul from 12–15 March for a non-leap year. Note that the model peak temperature was observed on January 15 (15th day of the year) in all three cities. Therefore, the lag between the time of disease incidence maximum and time of minimum temperature depended on the model and city and was 38−40 days in Chelyabinsk, 45−49 days in Yekaterinburg, and 56−59 days in Barnaul. Figure 6 shows the schematic model-predicted illustration of the average seasonal curves for RI rates each city. The seasonal curves of RI rates for the two neighboring cities, Chelyabinsk and Yekaterinburg, were very similar. The seasonal curve for Barnaul had a lower amplitude and peaked later than the two other locations. The seasonal curves for ambient temperature were identical for the three locations and, thus, in Figure 6, one superimposed seasonal curve for temperature averaged across three cities is shown. 

## 4. Discussion

In summary, we introduced a multi-step modeling procedure and illustrated its ability to define the cycle length or period, select the shape for a long-term trend, and perform peak identification. With the proposed approach we were able to reconstruct the daily incidence of infections in each city and demonstrate with high accuracy that, while the general seasonal shape of the RI incidence looked the same, each city had its own seasonal features. Despite the substantial distances between Chelyabinsk or Yekaterinburg, and Barnaul, the peak temperature was observed on the same date, 15 January, in all three cities. The peaks of RI ranged from mid-February to early April exhibiting a 9–16-day shift in peak incidence among the cities. Remarkably, the lag between the time of temperature minimum and the time of the rotavirus incidence maximum depended on the city and was approximately ~40 days in Chelyabinsk, ~47 days in Yekaterinburg, and ~58 days in Barnaul. After adjusting for seasonality, the contribution of the environmental temperature to the explained variance of the time series of daily RI rates was quite small for both immediate and lagged effects (0.01%–0.04% and 0.26%–0.5%, respectively). It is unlikely that these differences will be detected if one applies a standard approach of using monthly records and reporting a month with the highest counts or rates. 

These observations raise a fundamental question of why peaks occurred at certain times. Diseases do not read calendars and people are unlikely to choose a specific day to be sick. Thus, if in specific locations RI rates are peaking with a remarkable precision at a specific time every year, this consistency is important to note. The existing theories for RI seasonality stipulate that many socio-demographic and environmental factors contribute to the seasonal change in RI incidence, including the effects of vaccination [36], age-specific symptomatology [14], population composition and birth rates [18], genotypic profile of locally circulated strains [37], extreme environmental temperature on viral composition and stability [38], and the effects of overall climatic changes in continental climates [33]. Yet, the absolute and relative contributions of these factors are unknown and require further investigations. It is easy to speculate that the RI seasonality is complex and depends on many factors. However, it is not easy to find an explanation why it is complex and how those factors affect seasonality. At the moment, no studies yet offer a solid theory on disease seasonality. With this study, we had started with temperature as the most pronounced environmental indicator in cold climates, yet the suggested method has the strong potential to detect complex variations in a temporal behavior of a seasonal disease and consider more than one factor. 

The delays between a peak in rotaviral infection and a nadir in ambient temperature and precipitation were noted in many studies [7,8,9,10,11,14]. In the temperate climate of England, Wales, Scotland, and The Netherlands, the effect of temperature was delayed by up to four weeks for reported rotaviral infections [1]. For tropical climates, a meta-analysis of 26 studies conducted between 1975 and 2003 demonstrated a strong inverse relationship between monthly rotavirus incidence and climatological variables such as temperature, rainfall, and relative humidity [5]. In a recent study, we found significant correlation at a 2 month lag, indicating the delayed effects of cold and dry conditions on rotavirus incidence in Costa Rica [6]. Other studies have also reported negative lagged correlations between rotavirus and both temperature and precipitation in tropical and temperate regions of South Asia [3,13]. There is clearly an interest in identifying the order in which RI peaked in specific locations [9,12]. Yet, most of studies that attempted to estimate the lag between environmental parameters and RI or examine the sequence of RI peak timing opted out to use monthly counts or incidence, which substantially reduces the accuracy of estimation.

The peak timing along with their confidence intervals are rarely estimated with the sufficient precision to enable a formal comparison. The most common reporting of peak timing is a month with the highest number of cases or highest rates observed over multiple years. For example, a two-year multi-city study in Russia to characterize rotaviral infection among patients hospitalized with acute gastroenteritis suggested that the peak seasons of hospitalization due to the fact of RI were winter and spring, from December through May, based on a visual inspection [36]. When we applied the δ-method, we were able to estimate the range at which a peak occurs and to better characterize the shape of a seasonal curve. For instance, we have reported that, in Costa Rica, hospitalizations due to the presence of rotavirus in young children, on average, peak between late-February and early-March [6]. In India and Bangladesh, the primary peak in reported RI was observed in late-December to late-February with a smaller secondary peak in summer months of June–August [3]. By using daily records, we substantially increased the estimation accuracy. For example, we determined that in the United States, hospitalizations due to the fact of RI among older adults peaked on day 80 (95%; CI: 75–84) which is between 16–25 March [21] and which is in striking agreement with our findings. The daily records examined in this study represent all ages, yet RI is more common in children than in adults and the peak timing among age groups may vary, still mostly reflecting the younger population with its shift from symptomatic to non-symptomatic infection [37]. Unfortunately, the data we used in the study did not allow us to explore the difference in seasonal patterns across genetic profiles, which would allude to genome backgrounds circulated in cold climate of Russia and specifically in Siberia [38].

Growing utilization of electronic health records and improved laboratory testing foster the development of reliable surveillance systems to monitor local, regional, and global health and, thus, to improve resource allocation, health communication strategies, assessment of efficacy and efficiencies of public health intervention and prevention programs. By knowing when the infection may start to peak, public health authorities can exercise targeted health promotion campaigns, hospital managers can organize laboratory testing and treatment stockpiling, pharmaceutical companies can distribute vaccines, and physicians, nurses, and practitioners can be vigilant in detecting early cases. The production and distribution of vaccines, laboratory testing, and public health communication campaigns are costly and contribute substantially to the overall health care cost. The entire supply chain operation (e.g., which health care services are chosen, when and where they are delivered, how they are distributed) fails when a disease seasonal pattern is not taken into consideration or a disease spikes unexpectedly. By improving the accuracy of predicting seasonal peaks we could optimize prevention, health care delivery, and ultimately reduce costs. 

The applied approach has a strong potential in developing future forecasting models. Using the two-stage procedure we determined that the time series of daily RI rates in Yekaterinburg and Chelyabinsk have a similar temporal structure with the non-linear trend and complex seasonal components that describe the annual and semi-annual periods. The increasing rate of rotaviral infections has been depicted in all three cities, indicating the need for vigilant monitoring and the urgency for effective vaccination coverage. With the SSA model, we improved the accuracy of peak timing estimation and detected a potential secondary peak in summer months noted in the most recent years. While the contribution of the semi-annual period, corresponding to this summer peak, was not statistically significant (*p* > 0.05), it might be important in future forecasting models. We illustrated that polyharmonic modeling can be conducted in both complete and simplified forms (Models B and B’) and incorporates additional parameters (Models C and C’). Models C and C’ demonstrated the modest relative contribution of ambient temperature after adjusting for RI seasonality. The advantage of these models is that they could be further expanded to include other environmental factors. For instance, such expansions could incorporate covariates describing the joint effects of temperature and precipitation, important for tropical climates, or parameters reflecting growing population vaccination coverage, or the lag structure depicting the delayed effects of environmental parameters or population-wide disease control strategies [3,6,14].

In explored data, the formal tests rejected the Poisson assumption due to the disproportionally high contribution of days with low rates that is typical for infections with a well-defined seasonal pattern, e.g., long periods of low incidence alternated by a sharp rise and decline at regular intervals. The common approach for further improvement in model fitting would be to apply a negative binomial or a zero-inflated distribution. Although in our case such improvements would be negligibly small, yet this feature would be important in future forecasting models. We choose to illustrate the modeling approach using rotaviral infection for its notorious, globally observed, and well-defined winter peaks. The natural expansion of this work would be to apply the developed method to other winter infections, such as influenza, or summer infections, such as salmonellosis, and to incorporate it in the analysis of data collected by national and global surveillance systems. 

The temporal structure and seasonal peak estimates in a daily time series of cases of rotavirus were quite close based on two different approaches: a new class of spectral analysis model and a “traditional” harmonic model [23,24]. The new approach to detect and describe the periodic components is based on the use of singular spectrum analysis (SSA) and was implemented in an algorithm of identification and prediction of BP, the “Caterpillar” SSA [27]. One of the advantages of the “Caterpillar” SSA model is that the method has no requirement of a priori knowledge of the temporal structure governing the time series. In addition, the method more accurately reveals the temporal structure in the spread of a highly contagious infectious disease and takes into account the asymmetric nature of changes in the seasonal curve. The SSA model (Model A) informs polyharmonic models (Models B and C) on which temporal terms (trend and seasonal components) are essential. In our example, the SSA model more accurately describes the peak timing and explains most of the variance of the time series as compared to polyharmonic models. The main drawback of the method is the difficulty in analytical description and interpretation of the resulting model. Similar to machine learning methods, the basic SSA algorithm is a sequence of steps performed in an iterative manner, not a single formula. The shortcoming of the model is the lack of a collapsed parametric analytical description of the model, which does not allow testing of hypotheses about the presence in the time series of a component which complicates the interpretation of the model. This is one of the reasons we contrasted SSA with the polyharmonic regression which is now used broadly. In addition, the formation of the initial series of trajectory matrixes resulted in a loss of data (lost time equal to the length windows, in this case, 365 days), which increases the requirements for the length of the original time series, its observation period, and/or its measurement frequency. However, the framework of the traditional approaches offers a clear and meaningful interpretation of parameters and a set of characteristics that make it possible to describe the specifics of the seasonality of infectious disease. Further exploration of health records and environmental parameters compiled by the CliWaDIn database and similar sources will allow expansion and improvement of this methodology in understanding of the climate-health interactions [13,14,21,25,39].

The hybrid modeling based on simultaneous application of these approaches for the identification of temporal structure in infectious diseases is quite promising. Combined use of approaches makes it possible to design “quality” models of infectious diseases that can be used in practice, to describe and predict accurately the time of the peak of infectious diseases, their amplitude, and duration. This type of analysis enables comprehensive analyses of a temporal structure of infectious disease with complex pathways and environmental links and their seasonal characteristics. The ability to define and quantify seasonal peaks is essential in improving timely resource allocation, targeting health communication strategies, assessing the efficacy and efficiencies of public health intervention and prevention programs, and developing accurate forecasting of disease transmission. 

## 5. Conclusions

The two-step modeling procedure to examine seasonal peaks in rotaviral infection and the relationship between infection and ambient temperature allows to take advantage of the precision offered by singular spectrum analysis (SSA)—a new class of spectral analysis model—and the simplicity and interpretability of harmonic regression models. The ability to better quantify disease seasonality and its complex relationships with environmental parameters opens up new opportunities for accurate characterization of global disease patterns, optimizing resource allocation and health care delivery, understanding the contribution of factors governing multifaceted disease seasonality, and, eventually, for developing disease forecasting scenarios associated with prevention strategies and climate change.

## 6. Registration

The CliWaDIN database and codes for SSA algorithms are properly registered [29,35]. 

## Figures and Tables

**Figure 1 ijerph-16-04309-f001:**
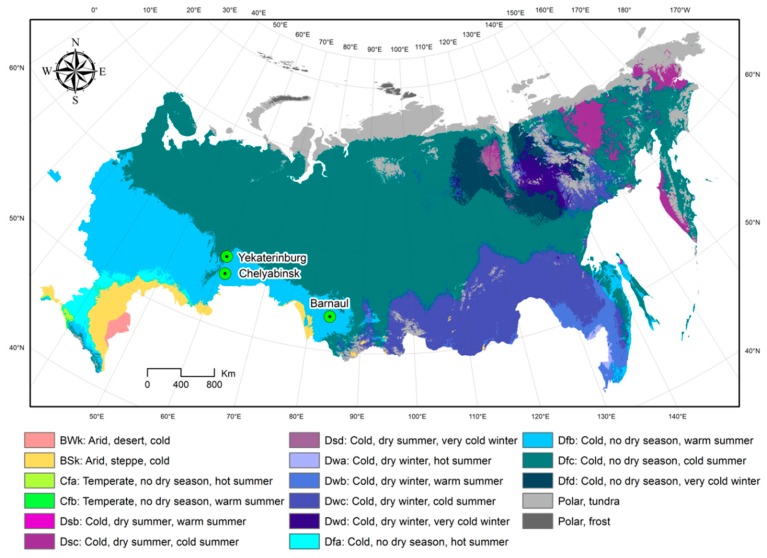
Köppen–Geiger climate classification and locations for three Russian cities: Chelyabinsk, Yekaterinburg, and Barnaul (adapted from Reference [33]).

**Figure 2 ijerph-16-04309-f002:**
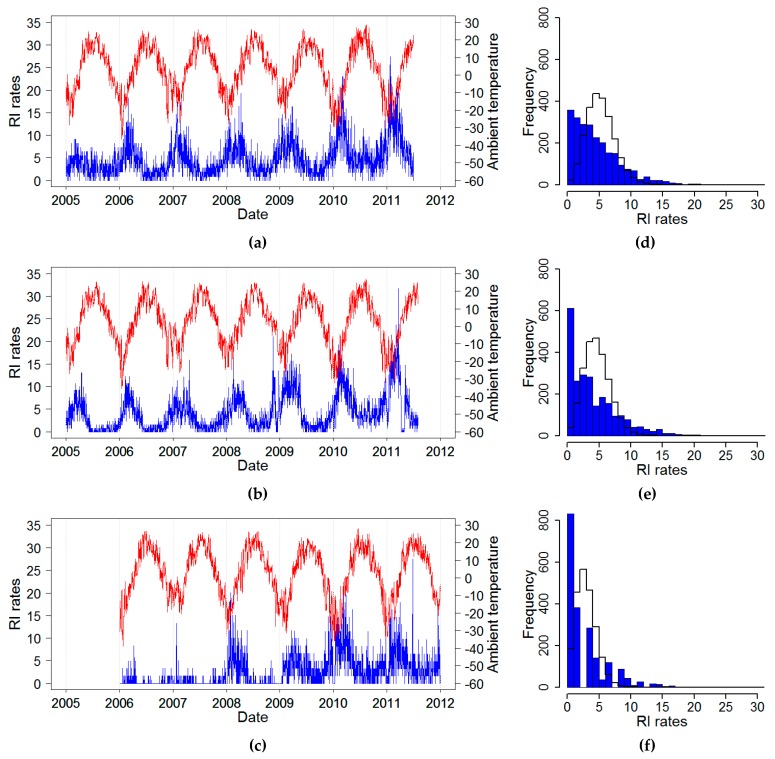
Time series of daily ambient temperature (red line) and daily rates of rotaviral infection (RI) (blue line) (**a**–**c**) along with the RI rate empirical distribution (blue color bars) and fitted Poisson distribution (black line) (**d**–**f**): (**a**,**d**) Chelyabinsk; (**b**,**e**) Yekaterinburg; and (**c**,**f**) Barnaul, Russia, 2005–2011.

**Figure 3 ijerph-16-04309-f003:**
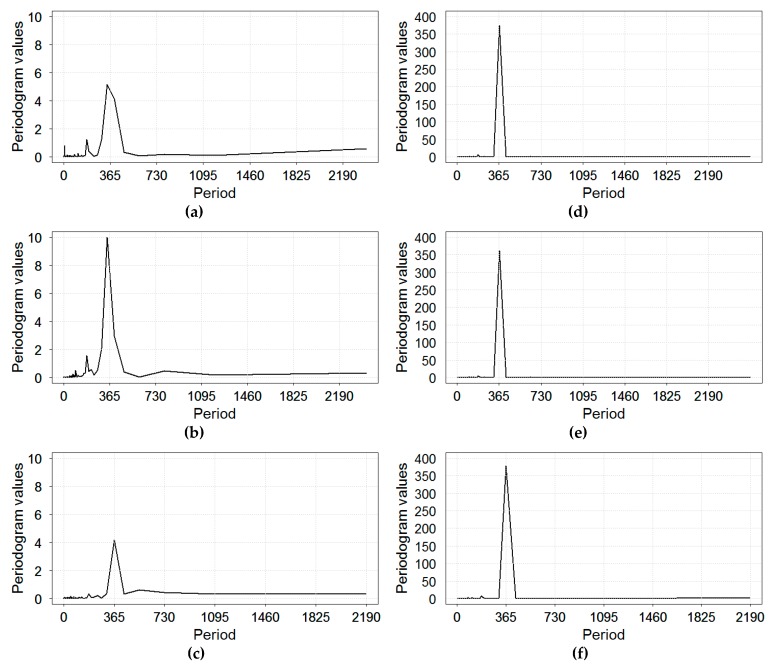
The periodogram spectral analysis for time series of daily rates of rotaviral infection (RI) (**a**–**c**) and daily ambient temperature (**d**–**f**) in three Russian cities: (**a**,**d**) Chelyabinsk; (**b**,**e**) Yekaterinburg; and (**c**,**f**) Barnaul, 2005–2011.

**Figure 4 ijerph-16-04309-f004:**
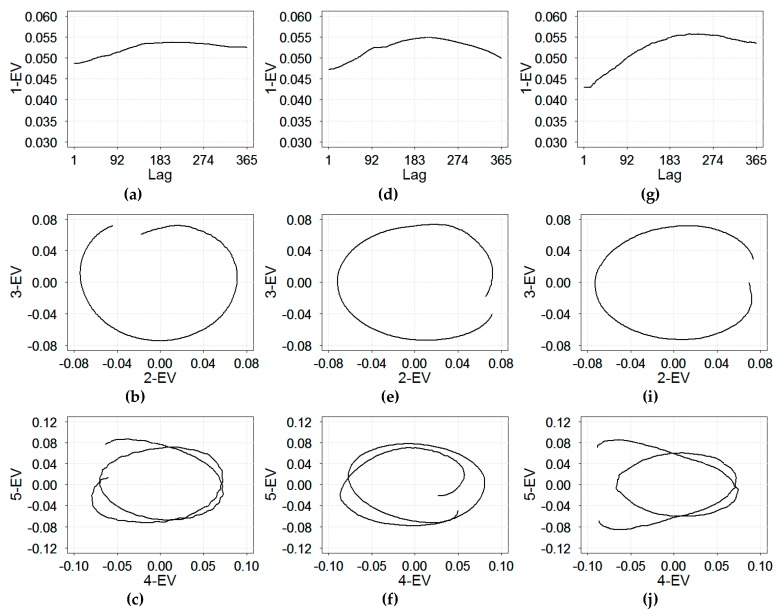
Plots of the eigenvectors (EV) pairs: 1-EV and lag, 2–EV and 3-EV, and 4-EV and 5-EV for Chelyabinsk (**a**–**c**); Yekaterinburg (**d**–**f**); and Barnaul (**g**–**j**).

**Figure 5 ijerph-16-04309-f005:**
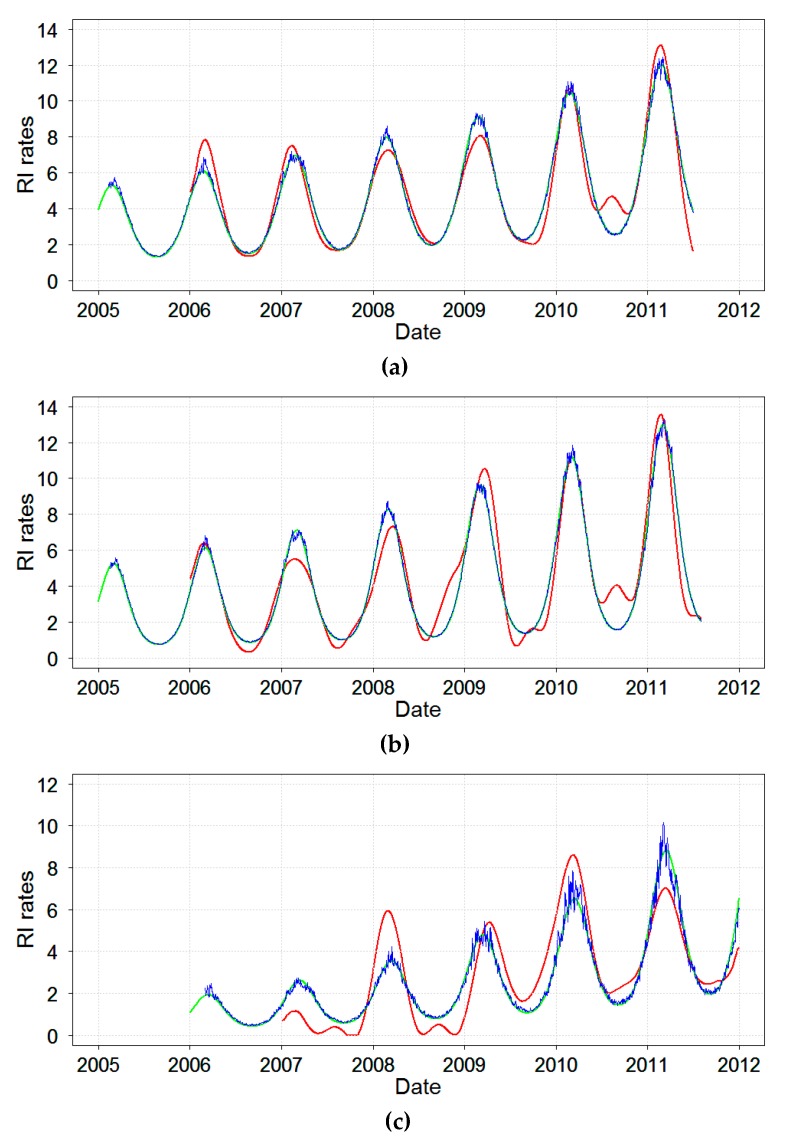
The predicted RI rates based on Model A (red line), Model B’ (green line), and Model C’ (blue line) in: (**a**) Chelyabinsk, (**b**) Yekaterinburg, and (**c**) Barnaul, Russia, 2005–2011.

**Figure 6 ijerph-16-04309-f006:**
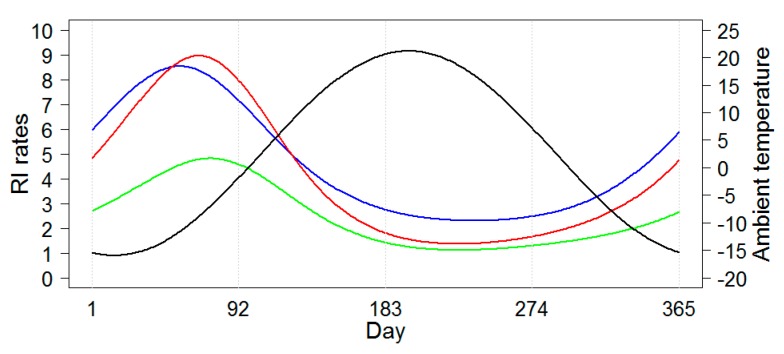
The average seasonal curves for daily time series rotaviral infection (RI) rate based on Model C’ for three cities, Chelyabinsk (blue line), Yekaterinburg (red line), and Barnaul (green line), and for the average daily temperature which was identical for all three cities (black line).

**Table 1 ijerph-16-04309-t001:** General study site characteristics: population, location, study period, and total number of rotavirus cases for three Russian cities: Chelyabinsk, Yekaterinburg and Barnaul, 2005–2011.

Parameter	Chelyabinsk	Yekaterinburg	Barnaul
General Characteristics:			
Population (mean)	1,103,862	1,326,133	607,785
2005	1,095,100	1,304,300	631,200
2006	1,093,000	1,308,400	604,200
2007	1,091,500	1,315,100	600,100
2008	1,092,500	1,323,000	597,200
2009	1,093,699	1,332,264	597,296
2010	1,130,132	1,349,772	612,401
2011	1,131,108	1,350,100	612,100
Latitude Longitude	55.1644° N 61.4368° E	56.8389° N 60.6057° E	53.3548° N 83.7698° E
Study period start Study period end	1 Jan 2005 30 Jun 2011	1 Jan 2005 31 Jul 2011	1 Jan 2006 31 Dec 2011
Number of days	2372	2403	2191
Number of cases	12,423	13,342	3307

**Table 2 ijerph-16-04309-t002:** Statistical characteristics of daily rates of rotaviral infection (RI) (outcome) and daily average temperature (exposure) for three Russian cities: Chelyabinsk, Yekaterinburg, and Barnaul, 2005–2011.

Parameter	Chelyabinsk	Yekaterinburg	Barnaul
*Outcome:* Daily RI rate per 1 M population			
Number of cases	12,423	13,342	3307
Mean ± SE	4.72 ± 0.08	4.16 ± 0.08	2.48 ± 0.07
1st; 3rd quartile	1.83; 6.40	0.77; 6.06	0; 3.34
Median; Maximum	3.66; 27.39	3.02; 31.71	1.63; 27.57
Skewness; Kurtosis	1.47; 3.07	1.43; 2.65	1.88; 4.46
*Exposure:* Daily ambient temperature in °C (Mean ± SE)			
Winter: Dec–Feb Spring: Mar–May Summer: Jun–Aug Fall: Sep–Nov	−13.27 ± 0.31 4.85 ± 0.35 17.91 ± 0.17 4.47 ± 0.35	−13.78 ± 0.32 3.93 ± 0.33 17.25 ± 0.18 3.55 ± 0.34	−15.84 ± 0.35 4.03 ± 0.41 18.02 ± 0.17 3.91 ± 0.37

**Table 3 ijerph-16-04309-t003:** Accuracy of model identification based on statistical characteristics of model residuals.

Cities Parameters	Models					
	A	B	B’	C	C’	D
Chelyabinsk						
Minimum	−7.715	−7.796	−7.107	−7.610	−7.178	−22.91
Mean	0.027	0.011	0.042	0.008	0.039	0.000
Median	−0.266	−0.317	−0.280	−0.310	−0.274	0.218
Maximum	15.42	15.97	16.58	16.28	16.82	16.24
Std. deviation	2.669	2.657	2.688	2.656	2.687	5.499
MAE *	1.986	2.013	2.029	2.012	2.032	4.317
D **	52.70	48.97	47.75	49.43	48.25	82.93
Yekaterinburg						
Minimum	−9.246	−10.57	−10.90	−10.71	−11.57	−21.43
Mean	0.027	−0.026	0.016	−0.027	0.010	0.000
Median	−0.180	−0.365	−0.258	−0.374	−0.276	0.301
Maximum	20.10	19.19	19.31	19.48	19.74	16.32
Std. deviation	2.475	2.531	2.575	2.533	2.582	5.561
MAE	1.737	1.778	1.789	1.779	1.795	4.369
D	62.77	58.85	57.45	59.39	57.80	82.36
Barnaul						
Minimum	−6.955	−8.232	−8.640	−8.621	−10.02	−23.15
Mean	0.003	−0.054	−0.086	−0.051	−0.091	0.000
Median	−0.264	−0.455	−0.680	−0.474	−0.692	0.124
Maximum	24.27	24.88	24.13	24.16	24.16	16.405
Std. deviation	2.678	2.626	2.728	2.625	2.741	5.954
MAE	1.810	1.774	1.914	1.761	1.929	4.707
D	43.06	40.31	35.54	40.84	35.84	82.94

* MAE: mean absolute error; ** D: variation explained (%).

**Table 4 ijerph-16-04309-t004:** Variance explained by the Model B and C’ components and the immediate and lagged effects of ambient temperature estimated from Model C’ in three cities.

Characteristics	Chelyabinsk	Yekaterinburg	Barnaul
Variance explained by the Model B by components:			
Trend	11.48	11.67	21.46
Annual seasonal component	36.73	46.49	18.51
Semi-annual seasonal component	0.759	0.690	0.343
Total variance	48.97	58.85	40.31
Variance explained by the Model C’ by components:			
Trend	10.35	10.47	15.03
Annual seasonal component	37.39	46.98	20.51
Temperature (immediate effect)	0.011	0.012	0.042
Temperature (lagged effect)	0.499	0.341	0.262
Total variance	48.25	57.80	35.84
Effect of ambient temperature:			
RR1	1.00063	1.00166	0.99517
LCI_RR1_ UCI_RR1_	0.99726 1.00401	0.99825 1.00509	0.98964 1.00072
RR2	0.99585	0.99608	0.98996
LCI_RR2_ UCI_RR2_	0.99276 0.99895	0.99309 0.99909	0.98510 0.99484

RR: relative risk; LCI and UCI: lower and upper bounds of the 95% confidence interval, respectively.

**Table 5 ijerph-16-04309-t005:** The estimates of peak timing in rotaviral infection for a non-leap year for models A, B’, and C’.

Cities Models	Chelyabinsk			Yekaterinburg			Barnaul		
	A	B’	C’	A	B’	C’	A	B’	C’
Peak *	55.17 (3.21)	55.14 (0.14)	53.16 (3.4)	64.17 (5.12)	62.14 (0.14)	60.43 (1.25)	71.11 (7.48)	74.17 (0.15)	72.83 (3.52)
Dates	24 Feb	24 Feb	22 Feb	5 Mar	3 Mar	1 Mar	12 Mar	15 Mar	14 Mar
Range (days)	42–62	55–56	40–65	53–80	62–63	56–70	56–99	74–75	63–84
Dates	11 Feb 3 Mar	24 Feb 25 Feb	9 Feb 6 Mar	22 Feb 21 Mar	3 Mar 4 Mar	25 Feb 11 Mar	25 Feb 9 Apr	15 Mar 16 Mar	4 Mar 25 Mar
Lag in days **	40	40	38	49	47	45	56	59	58

* Average peak (mean standard error). ** Average lag between time of disease incidence maximum and time of temperature minimum.

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
