# Peer review of "Rotavirus Seasonality: An Application of Singular Spectrum Analysis and Polyharmonic Modeling"

_ijerph, 2019, doi:10.3390/ijerph16224309_

Round 1
Reviewer 1 Report
Those will be sent as an email attachment since the electronic system does not accept the comments.

Author Response
We greatly appreciate Reviewer's comments and suggestions, which in our opinion allowed us to improve the clarity and readability, and expressed our heartfelt thanks in the Acknowledgments.
We highlighted extensive additions in red in the revised version.
Since this reviewer is not a biomathematician, no judgment was reached on the appropriateness of the proposed procedure (pages 4-7 of manuscript). It is, however, not obvious, whether the procedure is needed for the data of Fig. 2, panels a-c and the blue curves of panels of Fig. 2, panels d-f. The black lines of ‘fitted frequency distribution’ (Fig. 2, panels d-f) are not fully comprehensible.
Figure 2 illustrates the raw data, the subject of the analysis. We have provided an extensive description of Figure 2 in the text. The black lines show the ‘fitted frequency distribution’ and are essential as the proof that the constructed model is correctly specified. This step of verifying the empirical distribution is often omitted in epidemiological studies - when the form of distribution is assumed to be Normal or Poisson, but not verified. We are offering the best practice approach in this analysis.
Since one of the results of the bioinformatics analysis (Fig. 6) is very similar to the real data of Fig. 2, the usefulness of the proposed procedure is not fully apparent.
As Reviewer correctly pointed out we were able to find models that accurately represent the actual data. This was the goal and we had achieved the goal. We highlighted the usefulness of our work in the Discussion. The importance of our work is that we were able to reconstruct the daily incidence of infections in each city and demonstrate with high accuracy that while the general shape of the incidence looks the same, each city has its own seasonal features. Specifically, there were significant systematic differences in peak timing of infection in three cities. It is remarkable that the detected differences of 9-15 days were observed in RI infections while the nadir in temperature was identical in all three cities. It is unlikely that these differences will be detected if a standard approach of using monthly records was applied. These observations raise a fundamental question of why peaks occurred at certain times.
As it turns out in Discussion, the authors are aware of the multifaceted causes relevant for the phenotype RV seasonality.
We are agreed with Reviewer that the causes are multifaceted. However, the heterogeneity of multifaceted nature is too vague of an explanation. This is because one can expect that, in each of the cities, the population composition is unique as is the set of contributing factors for infection. Diseases do not read calendars and people are unlikely to choose a specific day to be sick. We had listed several factors that may contribute to seasonality.
In this reviewer’s view, if follows that a mathematical analysis of the seasonality of RV infections has to be based on many more data than just ambient temperature as used here. The analysis provided is considered to be incomplete and premature.
It is easy to speculate that the RI seasonality is complex and depends on many factors. However, it is not easy to find an explanation why it is complex and dependent on many factors. At the moment, no studies yet offer a solid theory on disease seasonality. We have bits of information of various qualities that suggest various contributors to RI seasonality, yet no one has assessed the relative contribution of each potential factor and ranked them to say which one is the least or most important, or if just temperature is sufficient to start. We strongly believe that the scientific methods require careful consideration of each potential contributor and we should start somewhere. So, we had started with temperature as the most pronounced environmental indicator in cold climates. We offered a novel application of a method with the strong potential to detect complex variations in a temporal behavior of a seasonal disease and consider more than one factor. We had provided a detailed description of the models, their justifications, metrics of model quality, interpretation of results and directions for future studies. Is not clear which part of the undertaken study is incomplete and premature.
Specific Comments
Page 1 Introduction line 1. Consider phrasing: … like influenza and RV-associated diarrhea…
As textbooks suggest that diarrhea is not a disease, but a symptom, we modified the sentence as follows: “Global surveillance for major infectious diseases, like influenza and rotaviral infection.”
Line 4 from bottom. The data of the cited papers (ref 1-5) do not all support seasonality of RV infection nor do classic papers on RV infection seasonality. Whereas seasonality in countries of temperate climates is generally accepted, the reports on seasonality of RV infections in tropical regions came to variable conclusions. Patel et al, 2013 [ref 5] conclude that ‘a single unifying explanation for variation in the seasonality of rotavirus disease is unlikely.’ Cao et al, 2019 (cited below) conclude that the RV epidemiology is complex and may differ from one site to another. Colston et al, 2019 (cited below) conclude that RV seasonality is multifactorial. Variability of transmission events and the socioeconomic condition of low income are also considered as a factor of seasonality (Colston et al, 2018; Pitzer et al, 2011 (cited below); Patel et al, 2013 [ref 5]
Brandt CD, Kim HW, Rodriguez WJ, Arrobio JO, Jeffries BC, Parrott RH. Rotavirus gastroenteritis and weather. J Clin Microbiol. 1982 Sep;16(3):478-82.
Konno T, Suzuki H, Katsushima N, Imai A, Tazawa F, Kutsuzawa T, Kitaoka S, Sakamoto M, Yazaki N, Ishida N. Influence of temperature and relative humidity on human rotavirus infection in Japan. J Infect Dis. 1983 Jan;147(1):125-8.
Armah GE, Mingle JA, Dodoo AK, Anyanful A, Antwi R, Commey J, Nkrumah FK. Seasonality of rotavirus infection in Ghana. Ann Trop Paediatr. 1994;14(3):223-9.
Koopmans M, Brown D. Seasonality and diversity of Group A rotaviruses in Europe. Acta Paediatr Suppl. 1999 Jan;88(426):14-9.
Pitzer VE, Viboud C, Lopman BA, Patel MM, Parashar UD, Grenfell BT. Influence of birth rates and transmission rates on the global seasonality of rotavirus incidence. J R Soc Interface. 2011 Nov 7;8(64):1584-93.
Sarkar R, Kang G, Naumova EN. Rotavirus seasonality and age effects in a birth cohort study of southern India. PLoS One. 2013 Aug 16;8(8):e71616.
Colston JM, Ahmed AMS, Soofi SB, Svensen E, Haque R, Shrestha J, Nshama R, Bhutta Z, Lima IFN, Samie A, Bodhidatta L, Lima AAM, Bessong P, Paredes Olortegui M, Turab A, Mohan VR, Moulton LH, Naumova EN, Kang G, Kosek MN; Mal-Ed network. Seasonality and within-subject clustering of rotavirus infections in an eight-site birth cohort study. Epidemiol Infect. 2018 Apr;146(6):688-697.
Colston JM, Ahmed AMS, Soofi SB, Svensen E, Haque R, Shrestha J, Nshama R, Bhutta Z, Lima IFN, Samie A, Bodhidatta L, Lima AAM, Bessong P, Paredes Olortegui M, Turab A, Mohan VR, Moulton LH, Naumova EN, Kang G, Kosek MN; Mal-Ed network. Seasonality and within-subject clustering of rotavirus infections in an eight-site birth cohort study. Epidemiol Infect. 2018 Apr;146(6):688-697.
Chao DL, Roose A, Roh M, Kotloff KL, Proctor JL. The seasonality of diarrheal pathogens: A retrospective study of seven sites over three years. PLoS Negl Trop Dis. 2019 Aug 15;13(8):e0007211.
We are thankful to Reviewer for providing references and had examined each suggested citation. While we were aware of these publications and had contributed to some, we did not cite as most of the studies used monthly counts, a crude way to measure seasonality; yet in absence of any other metrics, we are using monthly data as well and understand the limitations. Based on the Reviewer’s comments we had decided to add the citations and to emphasize potential deficiencies.
We had included the suggested citations in the discussion, although the paper we had cited (ref 1-5) fully supports the seasonality of RV infection, by stating that RI varies over a course of a calendar year with relatively short periods of high incidence alternating with prolonged periods of low incidence. Perhaps the definition of seasonality is not clear, so we had provided the definition in the text. We expanded the discussion section on the multifaceted nature of RI seasonality, although this was not the point. This paper offered a way to characterize seasonality, e.g. define when it peaks with respect to the most unifying global clock in temperate climates – an ambient temperature. We had highlighted this aspect of the discussion.
Curiously, I (Naumova EN) did not realize that two of my papers on rotavirus seasonality are considered classic and one I had cited is not.
Paragraph 2. The text should be shortened.
Reviewer 2 had asked to clarify what is “time series” and we had expanded this section per request.
Paragraph 3, line 1. Consider citation of: Jonesteller CL, Burnett E, Yen C, Tate JE, Parashar UD. Effectiveness of Rotavirus Vaccination: A Systematic Review of the First Decade of Global Postlicensure Data, 2006-2016. Clin Infect Dis. 2017 Sep 1;65(5):840-850.
Thanks, reference has been added.
Paragraph 3, line 2. Refs 1-6. Not all data support seasonality in tropical climates; see comment above.
We modified the sentence to support the assertion that seasonality is not always detected: “However, the seasonal pattern and strength of the association between the incidence of rotaviral infection and ambient temperature, and the sensitivity of a seasonal peak of rotavirus to temperature vary.”
Paragraph 3, line 5. The delay has been explained to some extent. See Pitzer et al, 2011, cited above.
The suggested paper focuses on explanation in disease seasonality not the delay with ambient temperature. Pitzer et al did not show the raw data but offered time series predicted by the model and stated that the model predicted quite well the marked seasonal variations. It did not reject the seasonal pattern but attempted to explain it with changes in birth rates. The statement provided in their abstract: “Our results suggest that the relative lack of rotavirus seasonality observed in many tropical countries may be due to the high birth rates and transmission rates typical of developing countries rather than being driven primarily by environmental conditions.” This might be confusing because the environmental conditions especially temperature vary little in tropics but could be taken literarily without understanding the results presented in Fig 3 with the relative size of peak seasonal incidence and weekly incidence. In other words, their piece describes and explains rotavirus seasonality but explores how its driving forces may be more due to demographic factors such as birth or transmission rates rather than environmental factors.
Paragraph 5. The justification of using SSA is not obvious.
We had added more justification for using SSA: “The classical SA could be applied to time series of disease incidence with the goal to find a recurrent periodic pattern, a pattern that repeats itself over time, for example, a summer increase in enteric infections observed every year. This method does not require a priori knowledge of the model structure.”
Paragraph 6. Use past tense throughout [… developed… identified… etc]
We corrected the use of the past tense.
Fig. 1. Explain the acronyms of the color codes.
We added clarifications for the letter code: “The Köppen climate classification divides climates into five main climate groups: A (tropical), B (dry), C (temperate), D (continental), and E (polar), with each group being divided based on seasonal precipitation and temperature patterns, also represented by a letter.”
Last line. Ref. [22] is followed by ref [27] on page 6, penultimate paragraph. From here onwards the citation of refs [23]ff is out of the order of first time of citation. For the whole manuscript the congruence of citation numbers in text and ref. list has to be thoroughly revised.
The reference list has been revised.
4 to page 7, top. This reviewer is not competent to judge the suitability of the method proposed. The expertise of a biomathematician should be consulted. The different PPHR models and their differences are not clearly explained.
We provided extensive clarifications.
7 Fig. 2. See General Comments.
8 Table 2. Not all parameters are explained.
We expanded the explanation of Table 2.
Fig. 3. It should be clarified which panels relate to RV infections and which to ambient temperature. The numbers at the abscissae appear to indicate days (cumulative). Please clarify.
The explanation has been provided.
9 Fig. 4. The eigenfactor curves require explanation.
The explanation has been provided.
10 Table 3. This table requires an explanatory footnote.
This table already has footnotes explaining MAE and D abbreviations, which are fully defined in the Methods section. The rest of the terms are self-explanatory. As stated in the title they are statistical characteristics (min, mean, median, max, and std deviation) of the residuals. We added the word “model’ in the title.
11 Table 4. See comment on Table 3.
We provided footnotes.
12 Fig. 5. It is not clear how the RV infection rates predicted according to the different PPHR models differ from the observed curve.
We added a clarifying sentence:” As expected, all models depicted the general seasonal nature of RI and predicted from 36% to 59% of variability.”
13 Table 5. See comment on Table 3.
This table already has footnotes explaining peak timing and lag; the dates and ranges are self-explanatory.
Fig. 6. It is not clear how the data shown differ from those of Fig. 2.
Figure 2 shows the actual daily records. Figure 6 shows the schematic model-predicted illustration of the average seasonal curves. Note Figure 2 shows 8 years of data and Figure 6 shows a typical seasonal curve for 356 days.
Discussion. The reviewer supports the notion that seasonality of RV infections is a multifactorial phenotype, in which ambient temperature, ambient humidity/rainfall (not considered), age-related symptomatology of RV infections (not considered), laboratory confirmation of RV infection (not considered), genotypes of locally co-circulating RV wildtype strains (not considered) come together and can differ from one region to another (certainly at the distances of the three Russian cities compared). RV vaccination could not have been considered yet but will influence seasonality [See: Tate JE, Panozzo CA, Payne DC, Patel MM, Cortese MM, Fowlkes AL, Parashar UD. Decline and change in seasonality of US rotavirus activity after the introduction of the rotavirus vaccine. Pediatrics. 2009 Aug;124(2):465-71].
We added the suggested citation and agree that in general RI seasonality could be affected by many factors as we had mentioned in the discussion and had suggested exploring in the future work considering the modeling approach we had proposed. Perhaps, this future study is likely to be very expensive. We had collected anonymized data for 29,072 laboratory-confirmed cases to fully represent over 3 million people. To add information on strain genotype for almost 29,000 samples is costly, and while it would offer some additional information on the changes associated with peak timing, it will still not explain why one strain peaked early and another later. With commonly used monthly counts, it is unlikely that the difference will be detected at all. In our own study (Sarkar R, Kang G, Naumova EN. Rotavirus seasonality and age effects in a birth cohort study of southern India. PLoS One. 2013 Aug 16;8(8):e71616) we demonstrated that even when one strain changed in peak timing, it still aligned closely with the time when other strains peaked.
Although we did not consider other weather-related parameters, their contributions are somewhat questionable. For example, ambient humidity/rainfall in cold climates have to be considered with caution. During winter when the temperature is below -10C most of the time, there is no rainfall. Inclusion of precipitation would come in the form of snowfall, which is difficult to measure systematically. Humidity reflects water content in the air masses, so when it is cold in cold climates, humidity is very low and highly correlated with low temperature. Therefore, when we considered the temperature in our model, we indirectly considered humidity as well.
The reviewer has stated that we did not consider laboratory confirmation of RV infection. This is incorrect. We had repeated throughout the paper that we used laboratory conformed cases of RI to estimate daily rates.
We recognize that most of the cases detected in this study are children, and it is possible that RI in children and adults may have peaked at different times. Yet, this difference can’t provide an explanation of why the difference exists. In other words, the association between infection rates and temperature can be measured across age groups but age does not dictate infection rates. Most importantly, age could not be detected with commonly used statistical models and roughly aggregated monthly records.
14 Paragraphs 1, 2. Here a few very reasonable arguments are made.
Paragraph 2, line 5. … based on visual inspection… [ref 23]. This is an important point (see comment p 13). During RV seasons in temperate climates not all children with acute gastroenteritis are diagnostically confirmed as infected with RV. Other enteric viruses (human caliciviruses, astroviruses, enteric adenoviruses, and others) can cause clinical symptoms very similar to those of RV infections.
We used laboratory conformed cases of RI, not diarrheal symptoms. Children are not always tested for RI when they have symptoms. RI is not always manifested in the same way either. These clinical shortcomings still could not explain why RI peaks in the winter, not in the summer.
Paragraph 4 to page15, paragraph 2. The justification for the new procedure is not very transparent.
We had provided the additional clarification: “Similar to machine learning methods, the basic SSA algorithm is a sequence of steps performed in an iterative manner, not a single formula. The shortcoming of the model SSA is the lack of a collapsed parametric analytical description of the model, which does not allow testing of hypotheses about the presence in the time series of a component, which complicates the interpretation of the model. This is one of the reasons we contrasted SSA with the polyharmonic regression, which now has been used broadly.”
15 The conclusion statement is over-optimistic. The seasonality of RV infections is a multifactorial trait that is not fully understood by bioinformatic analysis of only one or two factors. The authors do not indicate further research directions as promised on p.3, line 4.
Our conclusion statement is about the proposed statistical method that allows one to find the disease peak with the highest accuracy the data offers: “The two-step modeling procedure to examine seasonal peaks in rotaviral infection and the relationship between infection and ambient temperature allows to take the advantage of the precision offered by Singular Spectrum Analysis (SSA) - a new class of spectral analysis model - and the simplicity and interpretability of harmonic regression models.”
We modified the second sentence as: “The ability to better quantify disease seasonality and its complex relationships with environmental parameters open new opportunities for accurate characterization of global disease patterns, optimizing resource allocation and health care delivery, understanding the contribution of factors governing multifaceted disease seasonality and eventually for developing disease forecasting scenarios associated with prevention strategies and climate change.”
Suppl. Table S1. Provide an explanation of the numbers listed.
The explanation has been provided in the Methods:” The compiled data set containing daily time series of RI rates and temperature is provided in Supplemental Table S1.”
16f References
Ref [16] is incomplete.
We applied the book citation requirements.
Ref [18]. Has this work been published in a scientific journal?
Only partially and we cited the relevant publications.
Ref [19] is incomplete.
We applied registered database citation requirements.
Refs [20, 21]. Read: … available…
Corrected
Ref [24} is incomplete.
We applied registered database citation requirements.
Reviewer 2 Report
Alsova et al., proposed a new mathematical model application method that can predict the incidence of seasonal rotavirus infection. The research is prudence, and results show the Model C's numerical theory is useful, but the content of the paper needs to be improved to be more readable to the reader. Therefore, the following revision is proposed.
Abstrsct
“However, analytical tools suitable for identification of seasonal peaks are still limited.”
Please describe the need to predict seasonal peaks in detail. Who needs it most (country, pharmaceutical company, vaccine manufacturer, practitioner, mother, smartphone health app developer)?
“15th of January ”
Please state whether it is difficult to predict RI using only information that predicts the season of the lowest temperature of the year.
“The daily rates reached their seasonal peaks ~9–16 days”
Please rewrite the sentence in a way that is easy to understand.
“apart: on 55.17±3.21 day of the year in Chelyabinsk, 64.17±5.12 day in Yekaterinburg, and 71.11±7.48 day in Barnaul.”
The meaning of the front number is difficult to understand. Please add an example of the meaning of the front and rear numbers.
”38-40 days” “45-49 days” “56-59 days”
Please indicate which numerical formula these numbers were derived from.
“can be further improved”
Please indicate numerically the prediction accuracy using only temperature information has been improved for the SSA prediction accuracy.
Introduction
“δ-method”
Please describe the types of variables used in the δ method.
“classical spectral analysis (SA)”
Please describe the types of variables used in the classical spectral analysis (SA).
“Spectrum Analysis (SSA)” ”Poisson PolyHarmonic regression (PPHR) model”
Please describe the types of variables used in the Spectrum Analysis (SSA).
“build a parametric harmonic model, from which we determine and compare seasonal characteristics”
Please describe the type of variable used.
“daily rates of RI”
Please describe how these rates were examined and quantified.
Please describe the type of variable used.
“a daily time series of RI rates”
Please describe the difference from “daily rates of RI”
“a new class of spectral analysis model, fit PPHR models”
Please describe the type of variable used. Is it mathematically new or epidemiologically new?
2. Data and Methods
Table 1
“Number of days” “Number of cases”
The total population data for each city in each year were shown in Table 1. However, it is unclear how the “Number of days” and “Number of cases” data were calculated. The authors should add a detailed description of “Number of days” and “Number of cases”.
2.2. SSA Method
Knowledge about Basic SSA is also described in WIKIPEDIA (https://en.wikipedia.org/wiki/Singular_spectrum_analysis). Please add information about the RI epidemiology that can be assigned to the Basic SSA formula using statistical terms understood by nursing students.
2.3. Poisson PolyHarmonic Regression (PPHR) model
“The general form of a PPHR model is:??=???(?0+Σ(?????(2????)??=1+?????(2????))+??), (5)”
There is not enough explanation about PPHR model. The origin of this formula of PPHR model and the range of the author's arrangement is unclear.
“Using the results of SSA modeling, we build the PPHR for the daily rates with the following structure (Model B):”
A detailed explanation of how SSA modeling was used is required.
“The seasonal terms: ???(2???) and cos (2???) represent the annual oscillation; the seasonal terms: ???(4???) and ???(4???) represent the semi-annual oscillation; the function of trend is the third-degree polynomial.”
Please describe why ???(4???) and ???(4???) is required as formula assignment fields.
“we also explored a simplified version of Model B, Model B' with one linear trend term:”
Please describe the informatics characteristics of Model B' compared to Model B due to the input of the formula.
“including two terms for ambient temperature: one for an immediate effect with t-1 and the second term – for a lagged effect with t-l. Thus, Models C and C' have structure:”
Please add a description of the reason for adding the expression of "?1??−1+?2??−?" to the specified location in the formula. A more detailed explanation of “two terms for ambient temperature (?1??−1+?2??−?)” is required.
“we used a harmonic regression model with a Gaussian distribution for ambient temperature ?? as the outcome variable, Model D:”
Please add an explanation of what is "a harmonic regression model". Please add an explanation details about how Model D is to be created.
“?1,?1 – the estimated parameters of the model for two harmonics;”
The description is missing, please add a detailed description.
Figure 2
Please add the figure legend for Figure2 (d), (e), (f). "RI rates" should describe how it was calculated.
“although the Kolmogorov-Smirnov tests rejected the Poisson assumption due high contribution of days with low rates.”
Please describe the purpose of using "Kolmogorov-Smirnov tests”
Table 2
Please add the description of the parameters in Table 2.
Figure 3
There is not enough explanation in figure legend about the difference between the left panels (a) (b) (c) and the right panels (d) (e) (f).
Figure 4
“three types of components: trend, seasonal components, and noise (the remainder).”
There is the lack of explanation for these components.
There are not enough explanations of figure legends for 1-EV, 3-EV and 5-EV.
Table 4
Explanation about "RR1, LCI, UCI, RR2, LCI and UCI" are insufficient.
Discussion
The authors should actively discuss the results of this study. Regarding Figure 5, (1) Reasons for the waveform periods of Model A, Model B, and Model C to be similar; (2) Reasons for the Model B and Model C waveforms to closely approximate; (3) Pleases to describe the reason why it is easy to shift compared Model A waveform to Model B and Model C. Does the authors think that models A, B, and C has the same ability to predict when RI will occur?
Please describe why authors chose RI as the target disease for this study. Why didn't the authors pay attention to other seasonal infections such as influenza?
How were clinical data collected at medical facilities? What kind of statistical data is available for readers to apply the numerical theory of this paper?
Conclusions
What are the medical benefits of using the numerical theory introduced in this paper?
Author Response
We greatly appreciate Reviewer's comments and suggestions, which in our opinion allowed us to improve the clarity and readability, and expressed our heartfelt thanks in the Acknowledgments.
We highlighted extensive additions in red in the revised version.
Abstrsct
“However, analytical tools suitable for identification of seasonal peaks are still limited.”
Please describe the need to predict seasonal peaks in detail. Who needs it most (country, pharmaceutical company, vaccine manufacturer, practitioner, mother, smartphone health app developer)?
Thank you for the suggestion to expand on the value of understanding disease seasonal patterns. We had added the following paragraph to the discussion. “Growing utilization of electronic health records and improved laboratory testing foster the development of reliable surveillance systems to monitor local, regional and global health and thus to improve resource allocation, health communication strategies, assessment of efficacy and efficiencies of public health intervention and prevention programs. By knowing when the infection may start to peak public health authorities could exercise targeted health promotion campaigns, hospital managers organize laboratory testing and treatment stockpiling, pharmaceutical companies distribute vaccines, physicians, nurses, and practitioners be vigilant in detecting early cases. The production and distribution of vaccines, laboratory testing, and public health communication campaigns are costly and contribute substantially to the overall health care cost. The entire supply chain operation: which health care services are chosen, when and where they are delivered, how they are distributed, fails when a disease seasonal pattern is not taken into consideration or a disease spike unexpectedly. By improving the accuracy of predicting seasonal peaks we could optimize prevention, health care delivery and ultimately reduce costs.”
“15th of January ”
Please state whether it is difficult to predict RI using only information that predicts the season of the lowest temperature of the year.
We appreciate the suggestion and added a paragraph on why it is difficult to model seasonal variations in RI in the Introduction section. We also added a paragraph of why the observed peaks in RI are remarkable.
“The daily rates reached their seasonal peaks ~9–16 days”
Please rewrite the sentence in a way that is easy to understand.
We rephrased the sentence: “In SSA models, RI rates reached their seasonal peaks around February 24th, March 5th and March 12th (e.g. 55.17±3.21, 64.17±5.12, 71.11±7.48 day of the year) in Chelyabinsk, Yekaterinburg, and Barnaul, respectively.”
“apart: on 55.17±3.21 day of the year in Chelyabinsk, 64.17±5.12 day in Yekaterinburg, and 71.11±7.48 day in Barnaul.”
The meaning of the front number is difficult to understand. Please add an example of the meaning of the front and rear numbers.
The explanation provided “We estimated the average peak timing and its variation for RI rates and seasonal nadir for daily temperature based on each selected model.
”38-40 days” “45-49 days” “56-59 days”
Please indicate which numerical formula these numbers were derived from.
The explanation provided in the Methods section.
“can be further improved”
Please indicate numerically the prediction accuracy using only temperature information has been improved for the SSA prediction accuracy.
Explanation provided.
Introduction
“δ-method”
Please describe the types of variables used in the δ method.
We added clarifications:” The d-method converts the results of a harmonic regression model, specifically the regression coefficients of two periodic terms: sine and cosine that imitate seasonal variations, into time units, like days, weeks or months to accurately define when the disease incidence reaches its annual peak or nadir. This method also allows giving a sense of how much this peak timing may vary or how stable it is.”
“classical spectral analysis (SA)”
Please describe the types of variables used in the classical spectral analysis (SA).
We added clarifications: “The classical SA could be applied to time series of disease incidence with the goal to find a recurrent periodic pattern, a pattern that repeats itself over time, for example, a summer increase in enteric infections observed every year.”
“Spectrum Analysis (SSA)” ”Poisson PolyHarmonic regression (PPHR) model”
Please describe the types of variables used in the Spectrum Analysis (SSA).
Both “Spectrum Analysis (SSA)” and ”Poisson PolyHarmonic regression (PPHR) model” use daily rates of RI and ambient temperature.
“build a parametric harmonic model, from which we determine and compare seasonal characteristics”
Please describe the type of variable used.
We simplified the sentence: At the first step, we identify periodicity in the observed data and then, using this information, build models, from which we determine and compare seasonal characteristics. We then clarified: “We then applied PPHR models to examine the relationship between daily time series of RI and ambient temperature and estimate seasonal peaks.”
“daily rates of RI”
Please describe how these rates were examined and quantified.
Please describe the type of variable used.
We have clarified what is a daily time series as follow: ”If the incidence of infectious disease, observed in a specific population, is tracked in a systematic manner and is expressed in proper chronological order at the high temporal resolution, it allows to apply modern sophisticated methodology and conduct in-depth data analysis. In other words, if we assemble cases of health outcomes as they occur in time, aggregate these events with respect to time, say daily or weekly counts, and compile into so-called time series of case counts or rates, we will be able to apply analytical methods suitable for time series data. “
“a daily time series of RI rates”
Please describe the difference from “daily rates of RI”
The term “daily rates of RI” referred to “a daily time series of RI rates” and used interchangeably, similarly to “daily temperature” and “a daily time series of temperature”. We checked when such substitution is inappropriate and corrected when needed.
“a new class of spectral analysis model, fit PPHR models”
Please describe the type of variable used. Is it mathematically new or epidemiologically new?
We used daily rates of RI and ambient temperature and clarified that “To the best of our knowledge, the use of SSA model in the analysis of epidemiological data has been never done before.”
We had separated and clarified the senesces:” We identified the structure of temporal variation in daily rates of RI and ambient temperature using the SSA model - a new class of spectral analysis models. We then applied PPHR models to examine the relationship between RI and ambient temperature and estimate their seasonal peaks.
Data and Methods
Table 1
“Number of days” “Number of cases”
The total population data for each city in each year were shown in Table 1. However, it is unclear how the “Number of days” and “Number of cases” data were calculated. The authors should add a detailed description of “Number of days” and “Number of cases”.
The number of laboratory-confirmed cases of rotaviral infections is now added in the text (the first paragraph of Section 2. Data and Methods. The “Number of days” refers to the study period also provided in the same paragraph.
2.2. SSA Method
Knowledge about Basic SSA is also described in WIKIPEDIA (https://en.wikipedia.org/wiki/Singular_spectrum_analysis). Please add information about the RI epidemiology that can be assigned to the Basic SSA formula using statistical terms understood by nursing students.
We sympathize with the Reviewer's frustration with the SSA complexity. As we had stated in the Discussion section: “The main drawback of the method is the difficulty in analytical description and interpretation of the resulting model. The shortcoming of the model SSA is the lack of a collapsed parametric analytical description of the model.” Unfortunately, even Wikipedia did not explain this method in terms that it can be easily understood by nursing or graduate students in non-mathematical fields. As with any advanced method, the SSA is founded on a theory that is beyond introductory biostatistics we are teaching in graduate public health programs. This is one of the reasons we contrasted SSA with the polyharmonic regression, which now has been used broadly. Similar to machine learning methods, the basic SSA algorithm is a sequence of steps performed in an iterative manner, not a single formula. We added clarification to the text.
2.3. Poisson PolyHarmonic Regression (PPHR) model
“The general form of a PPHR model is:??=???(?0+Σ(?????(2????)??=1+?????(2????))+??), (5)”
There is not enough explanation about PPHR model. The origin of this formula of PPHR model and the range of the author's arrangement is unclear.
The description of all models components has been provided in the sentences immediately after the equation (5): “where is the outcome interest at time t; t=1, 2, …, N is time in days, where N is the number of days in a time series; is an intercept and the reflects mean daily values over the study period. The periodic components have a frequency of =1/365.25; are the estimated parameters of the model for k – number of the periodic components.” We added a sentence to clarify that we used the general form to develop a more tailored model: “From the general form (5) we select the components that are most reasonable to describe the annual and semi-annual oscillations based on the periodogram spectral analysis and the results of SSA modeling.”
“Using the results of SSA modeling, we build the PPHR for the daily rates with the following structure (Model B):”
A detailed explanation of how SSA modeling was used is required.
We had expanded the description as follow: “As shown in Figure 4 (a, d, g), the first eigenvector (1 EV) is changing slowly with respect to the lag; it describes the non-linear trend in RI rates, suggesting the need for considering additional trend terms in the harmonic regression models. Four other eigenvectors (2-5 EVs) exhibit regular periodic behavior, suggesting the need for seasonal terms. The second and third rows of Figure 4 show two-dimensional diagrams of the neighboring eigenvectors ordered by eigenvalues in ​​eigentriples. The relationship between the eigenvectors forms tight spirals with one (for 2-3 EVs) or two (4-5 EVs) coils. The number of coils indicates the period of a harmonic component. Therefore, the single coil for 2-3 EV components describes the annual periodicity equal to 365 days, warranting for an annual seasonal term in the harmonic regression models. The dual coils for 4-5 EVs components describe a six-month interval (182 days) oscillations in time series, suggestion additional term for semi-annual oscillations.”
“The seasonal terms: ???(2???) and cos (2???) represent the annual oscillation; the seasonal terms: ???(4???) and ???(4???) represent the semi-annual oscillation; the function of trend is the third-degree polynomial.” Please describe why ???(4???) and ???(4???) is required as formula assignment fields.
We clarified the sentence as follow: “The periodic terms: and represent the annual oscillation (when one seasonal peak is expected); the periodic terms: and represent the semi-annual oscillation (when two seasonal peaks are likely);”
The need for these terms has been confirmed by the analysis shown in Figure 3 and Figure 4.
“we also explored a simplified version of Model B, Model B' with one linear trend term:”
Please describe the informatics characteristics of Model B' compared to Model B due to the input of the formula.
We clarified that the simplified Model B’ has one seasonal term and one linear trend term as compared to Model B which has two seasonal terms and three terms for trend. We also added a sentence on why we develop a simplified version: We compared Model B and B' to understand the improvements in model accuracy by considering additional terms.
“including two terms for ambient temperature: one for an immediate effect with t-1 and the second term – for a lagged effect with t-l. Thus, Models C and C' have structure:”
Please add a description of the reason for adding the expression of "?1??−1+?2??−?" to the specified location in the formula. A more detailed explanation of “two terms for ambient temperature (?1??−1+?2??−?)” is required.
We had provided clarifications: ‘we expanded PPHR model (Models B and B') by including two terms for ambient temperature: one term ( for an immediate effect with 1-day lag (t-1) and the second term ( – for a lagged effect with a variable lag (t-l).”
We also added a clarification: “The inclusion of two terms for temperature is justified by a potential difference between a seasonal peak in RI and a seasonal nadir in temperature. If such difference is detected the two terms should reflect the immediate and the lagged effects.”
“we used a harmonic regression model with a Gaussian distribution for ambient temperature ?? as the outcome variable, Model D:”
Please add an explanation of what is "a harmonic regression model". Please add an explanation details about how Model D is to be created.
We added an explanation in the Introduction we defined harmonic regression models as “models that explicitly use sine and cosine periodic functions to imitate seasonal variations in the time series of disease incidence. The results of the harmonic regression model produce the estimates of parameters for sine and cosine terms with proper specification; and if any of the two parameters are statistically significant, one may suspect that the temporal variations are periodic within a one-year cycle, or seasonal.” We also provided clarification that: “In cold climates, on average ambient temperature exhibit only one seasonal peak and one season nadir, thus two harmonic terms should be sufficient.”
“?1,?1 – the estimated parameters of the model for two harmonics;”
The description is missing, please add a detailed description.
We used the explanation consistent throughout the whole Data and Methods Section, ? and ? - are the estimated parameters of the model. In other words, they are the regression coefficients, so we added this clarification.
Figure 2
Please add the figure legend for Figure2 (d), (e), (f). "RI rates" should describe how it was calculated.
The legends for Figure 2 (d), (e), (f) are already in the text: “RI rate empirical (blue color) and fitted (black line) distributions”. The firth paragraph of Data and Methods provides a detailed description of how the rates were calculated. We clarified as: “(estimated from reported daily counts and interpolated city population)”.
“although the Kolmogorov-Smirnov tests rejected the Poisson assumption due high contribution of days with low rates.”
Please describe the purpose of using "Kolmogorov-Smirnov tests”
As described in the first sentence of Section 2.1. Exploratory Analysis: “We examined the suitability of a Poisson distribution for daily RI rates based on distributional shape, coefficients of skewness and kurtosis, and Kolmogorov-Smirnov test.”
Table 2
Please add the description of the parameters in Table 2.
We had provided an additional description of Table 2.
Figure 3
There is not enough explanation in figure legend about the difference between the left panels (a) (b) (c) and the right panels (d) (e) (f).
We had provided a better explanation of labeling.
Figure 4
“three types of components: trend, seasonal components, and noise (the remainder).”
There is a lack of explanation for these components.
There are not enough explanations of figure legends for 1-EV, 3-EV and 5-EV.
We provided expensive clarification of Figure 4.
Table 4
Explanation about "RR1, LCI, UCI, RR2, LCI and UCI" are insufficient.
We provided some clarification in the text of the Results and Methods sections.
Discussion
The authors should actively discuss the results of this study. Regarding Figure 5, (1) Reasons for the waveform periods of Model A, Model B, and Model C to be similar; (2) Reasons for the Model B and Model C waveforms to closely approximate; (3) Pleases to describe the reason why it is easy to shift compared Model A waveform to Model B and Model C. Does the authors think that models A, B, and C has the same ability to predict when RI will occur?
We are thankful to Reviewer for the suggestions and added sentences in several places, including: “We had illustrated the polyharmonic modeling can be conducted in the complete and simplified forms (Models B and B’) and incorporates environmental parameters (Models C and C’) that could be expanded to include other factors.” We clarified that Model A informs which seasonal terms to include in models B and C.
Please describe why authors chose RI as the target disease for this study. Why didn't the authors pay attention to other seasonal infections such as influenza? How were clinical data collected at medical facilities? What kind of statistical data is available for readers to apply the numerical theory of this paper?
We are thankful to Reviewer for this suggestion and added the following text: “We choose to illustrate the modeling approach using rotaviral infection for its notorious, globally observed, and well-defined winter peaks. The natural expansion of this work would be to apply the methods to other winter infections, like influenza, or summer infections, like salmonellosis, and to incorporate it in the analysis of data collected by national and global surveillance systems.
Conclusions
What are the medical benefits of using the numerical theory introduced in this paper?
We are thankful to Reviewer for their attention to this issue and modified the sentence as follow: “The ability to better quantify disease seasonality and its complex relationships with environmental parameters open new opportunities for accurate characterization of global disease patterns, optimizing resource allocation and health care delivery, understanding the contribution of factors governing multifaceted disease seasonality and eventually for developing disease forecasting scenarios associated with prevention strategies and climate change.”
Round 2
Reviewer 1 Report
Rotavirus seasonality: an application of singular spectrum analysis and polyharmonic modelling
By Olga K Alsova et al (Corresponding author: Elena N Naumova)
Submitted to Intern J Environ Res Publ Health (Editorial No. IJERPH-590619 R1)
General Comments
This is the revised version of a manuscript the original of which has been studied and commented upon by this reviewer. The authors have carefully considered the comments and made changes which increase the transparency of the results reported. As stated earlier, this reviewer is not a bio-mathematician, and therefore assessment of the methods used (p5-8 of revised manuscript) should be carried out by (an) alternative reviewer(s). While it is now more apparent that daily measurements (Suppl. Table 1) provide a higher degree of accuracy, in Discussion more weight should be given to thoughts of how additional factors determining the incidence of rotavirus (RV) infection/disease in defined geographical areas could be accommodated.
Specific Comments
Page
9 Fig. 2. The legend with regard to panels d, e, f can be improved.
14 Fig. 5. The different models show close proximity of models B’ and C’, as compared to model A. Please express a logically based preference for one of the models.
15 Fig. 6. The Barnaul region is further south than the other two regions (Fig. 1). Why is the ambient temperature curve of Barnaul flatter and lower than those of the other two regions?
17 Paragraph 3. It should be considered how additional factors determining seasonal appearance of rotavirus infection/disease may influence or be incorporated into the models investigated so far.
Author Response
While it is now more apparent that daily measurements (Suppl. Table 1) provide a higher degree of accuracy, in Discussion more weight should be given to thoughts of how additional factors determining the incidence of rotavirus (RV) infection/disease in defined geographical areas could be accommodated.
We are thankful to the Reviewer for thoughtful comments and suggestions. We highlighted all the relevant changes in the text.
Specific Comments
Page
9 Fig. 2. The legend with regard to panels d, e, f can be improved.
We added clarifying text to figure legend: "the RI rate empirical distribution (blue color bars) and fitted Poisson distribution (black line)"
14 Fig. 5. The different models show close proximity of models B’ and C’, as compared to model A. Please express a logically based preference for one of the models.
We clarified the text as follows: "Model A obtained via SSA modeling provided the most refined structure of the studied time series of RI rates. In addition to a well-pronounced winter peak, Model A revealed irregularities and unusual summertime peaks. As expected, the two harmonic regression models, Models B’ and C’ have identified a temporal structure with one seasonal peak for daily RI rates in three cities similarly to Model A. Because Model C’ included a term for ambient temperature, it depicted variations in RI rates due to ambient temperature, mostly noted when RI rate reached its peaks. Thus, Model C’ demonstrated the modest contribution of temperature in predicting RI rates."
15 Fig. 6. The Barnaul region is further south than the other two regions (Fig. 1). Why is the ambient temperature curve of Barnaul flatter and lower than those of the other two regions?
We clarified the text as follows: "The seasonal curves of RI rates for two neighboring cities Chelyabinsk and Yekaterinburg are very similar. The seasonal curve for Barnaul had a lower amplitude and peaked later that two other locations. The seasonal curves for ambient temperature were identical for three locations, and thus in Figure 6 one superimposed seasonal curve for temperature averaged across three cities is shown.
17 Paragraph 3. It should be considered how additional factors determining seasonal appearance of rotavirus infection/disease may influence or be incorporated into the models investigated so far.
We expanded the paragraph as follows: "The increasing rate of rotaviral infections has been depicted in all three cities, indicating the need for vigilant monitoring and the urgency for effective vaccination coverage. With SSA-model we improved the accuracy of peak timing estimation and detected a potential secondary peak in summer months noted in the most recent years. While the contribution of the semi-annual period, corresponding to this summer peak, was not statistically significant (p>0.05), it might be important in future forecasting models. We had illustrated the polyharmonic modeling can be conducted in both complete and simplified forms (Models B and B’) and incorporates additional parameters (Models C and C’). Models C and C’ demonstrated the modest relative contribution of ambient temperature after adjusting for RI seasonality. The advantage of these models is that they could be further expanded to include other environmental factors. For instance, such expansions could incorporate covariates describing the joint effects of temperature and precipitation, important for tropical climates, or parameters reflecting growing population vaccination coverage, or the lag structure depicting the delayed effects of environmental parameters or population-wide disease control strategies [3,6,14]."